# Selective nucleophilic α-C alkylation of phenols with alcohols via Ti=Cα intermediate on anatase TiO2 surface

Xinze Du [1,2], Hongjun Fan [3] ✉, Shenglin Liu[1] & Z. Conrad Zhang[1,4] ✉

C−C bond forming reaction by alkylation of aryl rings is a main pillar of chemistry in the production of broad portfolios of chemical products. The dominant mechanism proceeds via electrophilic substitution of secondary and tertiary carbocations over acid catalysts, forming multiple aryl alkylation products non-selectively through all secondary and tertiary carbons in the alkyl chains but producing little α-C alkylation products because primary carbocations are poorly stable. Herein, we report that anatase TiO2 (TiO2-A) catalyzes nucleophilic α-C alkylation of phenols with alcohols in high selectivity to simply linear alkylphenols. Experimental and computational studies reveal the formation of Ti=C− bond with the α-carbon of the alkyl group at oxygen vacancies of the TiO2-A surface. The subsequent α-C alkylation by selective substitution of phenol *ortho*-C−H bond is verified by deuterium exchanged substrate and DFT calculations.

Catalytically directed specific carbon-carbon bond formation is at the forefront of modern organic synthesis[1,2] and broadly involves acid-base and metalation catalysis[3–9]. Alkylation processes have been broadly employed in chemical and fuel productions as the prevailing C−C bond-forming technologies[10], using Lewis or Brønsted acidic catalysts including mineral acids and acidic oxides such as aluminum oxide and zeolites[11–17].

Acid-catalyzed alkylation of aromatic compounds using unsaturated aliphatic hydrocarbons commonly proceeds via electrophilic substitution on aryl carbons by secondary or tertiary carbocations[18]. The carbocation mechanism yields very little primary carbon alkylated products because a primary carbocation rapidly rearranges to more stable secondary carbocations. Therefore, alkylation with long chain α-alkenes or primary alcohols typically produces a mixture of multiple branched alkyl products on an aryl ring through the formation $C_{aryl}$−$C_{alkyl}$ bond in which the $C_{alkyl}$ may be a secondary or tertiary carbon of the alkyl chain but the primary carbons are not favored. Therefore, selectively making superlinear α-alkylated aromatics without using pre-functionalized substrates has been a persistent challenge

in chemistry[14]. In addition, the aryl ring of alkylated products is often more kinetically reactive than the starting aromatic substrates, due to the electron-donating effect of alkyl groups. Therefore, monoalkylation is often not possible in high yield[19].

Phenolics are commodity chemicals. Recent works show that phenol can be produced from guaiacol, a pyrolysis product of lignocellulosic biomass[20–23]. About 540,000 tons of higher alkylphenols are synthesized worldwide per year[24]. Linear alkyl phenolics are important chemicals as intermediates in the agrochemical and surfactant industries[24]. Moreover, higher alkylphenols are monomers for various phenolic resins production[25]. Typical acid-catalyzed electrophilic substitution on phenol yields a mixture of *ortho*- and *para*-substituted phenol derivatives, with branched alkyls dictated by the characteristic distribution of more stable carbocation intermediates[26]. Fatty alcohols derived from animal fats and plant oils are abundantly available primary alcohols of linear alkyl chains[27–29]. Direct α-C alkylation producing pure superlinear alkyl phenol with high selectivity using metal-free inexpensive heterogeneous catalysts would be a highly desirable strategy to break the

[1]State Key Laboratory of Catalysis, Dalian National Laboratory for Clean Energy, Dalian Institute of Chemical Physics, Chinese Academy of Sciences, Dalian 116023, China. [2]University of Chinese Academy of Sciences, Beijing 100049, China. [3]State Key Laboratory of Molecular Reaction Dynamics, Dalian National Laboratory for Clean Energy, Dalian Institute of Chemical Physics, Chinese Academy of Sciences, Dalian 116023, China. [4]Changzhou University, Changzhou 213164, China. ✉e-mail: fanhj@dicp.ac.cn; zczhang@yahoo.com

limitation of existing processes, but remains technically elusive thus far.

The current route for the preparation of superlinear alkyl aromatics involves a Friedel–Crafts acylation followed by a Clemmensen reduction, which is not yet viable for large-scale processes[15,18]. To achieve site-selective *ortho*-alkylation of phenols, a strategy of modifying the hydroxyl group with a directing group is widely utilized[26,30]. However, the directing group must be pre-synthesized and then removed afterward[31–34]. Alkylation of phenol by a noble metal catalyst, e.g. Pd/C, in the presence of BuOLi base in a solvent was reported to produce *ortho*-selective α-carbon substitution[35]. And noble metal complex was also reported to be an alternative catalyst[36]. Base metal oxides can catalyze some specific alkylation reactions, for example, methylation of phenol to *o*-cresol with methanol in industry[37–40]. Further studies found that on the base sites, methanol was transformed to formaldehyde and reacted with adsorbed phenolate species to generate salicylic alcohol via hydroxymethylation at the primary stage[41]. Selective alkylation of phenol with 1-propanol was reported to produce 2-n-propylphenol over a CeO$_2$-MgO at 475 °C, and the mechanism was speculated via a radical process[42]. Like other refractory metal oxides such as Al$_2$O$_3$ and SiO$_2$, TiO$_2$ can be prepared by specific method to bear strong Lewis or Brønsted-acid sites[43,44] to catalyze the Friedel-Crafts alkylation of phenolics and arenes[43–46].

In this work, we report that catalyzed by anatase TiO$_2$, alkylation reactions of phenols with alcohols exhibit high selectivity to α-C alkylation products (α-C means the α-carbon in alcohol). According to the experimental results and computational studies, alcohol is activated at oxygen vacancies of the TiO$_2$-A surface to produce an alkyl group, which interacts with the Ti atom in the form of Ti=C$^-$ bond. The subsequent α-C alkylation to aromatic C−H bond is simulated by DFT calculations and verified by deuterium exchanged substrate experiment.

## Results

### Catalytic performances

Figure 1 shows the overall conversion rate of phenol and the yield of each product at 300 °C using H-ZSM-5, γ-Al$_2$O$_3$, rutile titania (TiO$_2$-R), P25 TiO$_2$ and anatase titania (TiO$_2$-A) to catalyze the model reaction, alkylation of phenol with 1-propanol. The catalytic performances of the two typical solid acid catalysts, H-ZSM-5 and γ-Al$_2$O$_3$, are consistent with the typical electrophilic mechanism as reported[47,48]. Apart from low 2-n-propylphenol yield, 2-isopropylphenol was formed as the dominant product. Propyl phenyl ether was generated from the intermolecular dehydration reaction of phenol with

1-propanol. Several isopropyl polyalkylation products are identified by GC-MS and calculated by NMR spectra with the internal standard method. Polyalkylation is common in acid-catalyzed alkylation because of the electron-donating effect of alkyl groups. On the other hand, 1-propanol was mostly consumed on the two solid acid catalysts (67.2% on H-ZSM-5 and 77.6% on γ-Al$_2$O$_3$), producing propylene and propyl ether by dehydration process (Supplementary Table 1, entries 1 and 2). Among several TiO$_2$ catalysts, rutile TiO$_2$ (TiO$_2$-R) was not active to catalyze the reaction and P25 TiO$_2$, which is a mix of TiO$_2$-R and TiO$_2$-A, showed a low conversion rate. However, TiO$_2$-A is clearly distinguished over the known acid-type catalysts. The phenol conversion rate reached 86.1%, with 74.9% 2-n-propylphenol yield and only 2.6% 2-isopropylphenol, corresponding to an 89.2% selectivity of n-propyl products on TiO$_2$-A. Meanwhile, the consumption of 1-propanol (27.1%) and yield of by-product propylene (3.6%) on TiO$_2$-A was significantly less than that on H-ZSM-5 and γ-Al$_2$O$_3$, indicating the efficient utilization of alcohol as alkylation reactant (Supplementary Table 1, entry 3).

We further verified this distinctively appealing catalytic nature of TiO$_2$-A with several alcohols and substituted phenols (Fig. 2). When phenol was used to react with 1-dodecanol, a lower conversion (67.0%) was obtained compared to 1-propanol under the same condition, but a high selectivity (78.2%) to superlinear alkylation product still dominated. The conversion of 1-dodecanol was 20.8%, with a small amount of 1-dodecene (2.7%) as by-product (Supplementary Table 1, entry 4). In addition, secondary alcohols, such as 2-propanol and 2-dodecanol, were also evaluated for alkylation of phenol (Fig. 2, entries 3 and 4). The TiO$_2$-A catalyst exhibited similar reactivity to the corresponding primary alcohols, and more importantly, maintained the selectivity of α-C alkylation. The reactions of 3-methylphenol and 3-chlorophenol as substituted phenols with 1-propanol were also investigated (Fig. 2, entries 5 and 6). The two reactions both generated *ortho*-substituted n-propyl products with more than 80% selectivity. It is worth noting that the conversions of these two phenols showed no obvious difference (94.3% and 90.7%, respectively), although the methyl group and chlorine group have opposite electronic effects as substituent groups. These results clearly indicate that the main alkylation mechanism on TiO$_2$-A surface is distinguishably different from the Friedel-Crafts alkylation. Phenols with additional substituents were further tested by reacting with 1-propanol (Supplementary Fig. 2, entries 1–6), and high conversions of phenols (more than 84%) and selectivity to *ortho*-substituted n-propyl products (more than 80%) were obtained. In

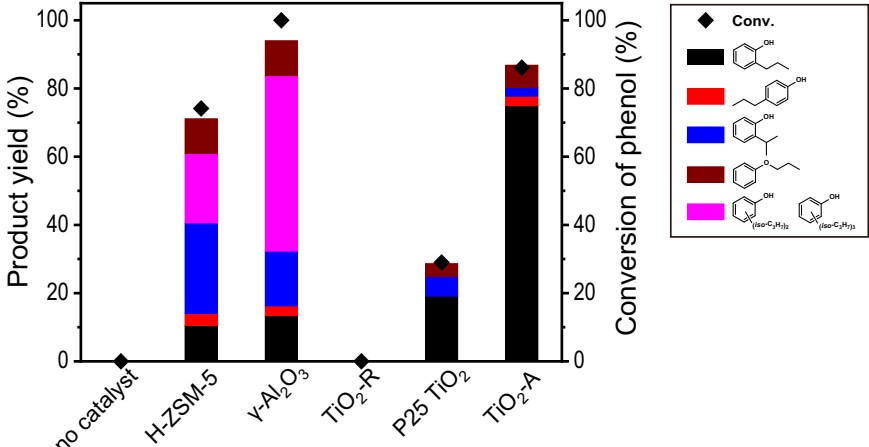

**Fig. 1 | Product yields in the alkylation of phenol with 1-propanol over different oxide catalysts.** Reaction conditions: catalyst 0.2 g, phenol 2.5 mmol, 1-propanol 10 mmol, toluene 25 mL, 300 °C, 16 h, N$_2$ atmosphere.

**Fig. 2 | The α-C alkylation of phenols with alcohols catalyzed by TiO₂-A.** Reaction conditions: TiO₂-A 0.2 g, phenol 2.5 mmol, alcohol 10 mmol, toluene 25 mL, 300 °C, 16 h, N₂ atmosphere.

addition, naphthols as alternative substrates also showed similar alkylation selectivity in high conversions (Supplementary Fig. 2, entries 7–10). These results demonstrate that the TiO₂-A catalyzed selective α-C alkylation reaction applies to a broad scope of substrates.

The universality of the reaction was also studied by the model reaction between phenol with 1-propanol. The conversion of phenol slightly varied in different solvent systems, while 2-n-propylphenol was invariably the main product (Supplementary Fig. 1a). When the reaction time was extended from 4 h to 16 h at 300 °C, the *ortho*-substituted n-propyl product selectivity gradually increased with time (Supplementary Fig. 1b), indicating that the formation of 2-n-propylphenol on TiO₂-A prevailed as the dominantly main catalytic mechanism and the sites that catalyze side reactions and 1-propanol consumption were suppressed. With the feeding ratio between phenol and 1-propanol varying from 1:1 to 1:4, product selectivity had no obvious changes while conversions of phenol gradually increased (Supplementary Fig. 1c) as a relatively high concentration of 1-propanol favorably drives the reaction equilibrium to the product.

## Characterization of TiO₂-A

To study the TiO₂-A catalyst for its unique capability of forming α-C alkylation, we performed characterizations of TiO₂-A before and after the reaction. Figure 3a–c shows the TEM images of TiO₂-A and used TiO₂-A. The TiO₂-A is composed of irregular nanoparticles with a diameter of about 10 nm. The observed interplanar spacing is 0.35 nm, corresponding to the (101) crystal plane of anatase TiO₂. The microstructure has no obvious change after the reaction. Figure 3d shows the XRD spectra of TiO₂-A before and after the reaction. These two spectra are both consistent with the anatase TiO₂ pattern (PDF #21-1272). X-ray photoelectron spectra of the catalyst (Supplementary Fig. 4) show that the valence state of Ti is +4, and the valence state of O can be divided into two parts, lattice oxygen and hydroxyl oxygen on the surface. In addition, the specific surface area of catalyst is 146 m²/g and 141 m²/g after reaction. The TiO₂-A catalyst composed of nanoparticles maintained its original structure after the reaction. Its structure stability was maintained with multiple reuse tests. After three-times reuse of the TiO₂-A, the phenol conversion and product yields remained no obvious changes (Supplementary Fig. 3a). It is interesting to note that the used TiO₂-A had a better mass balance than the fresh

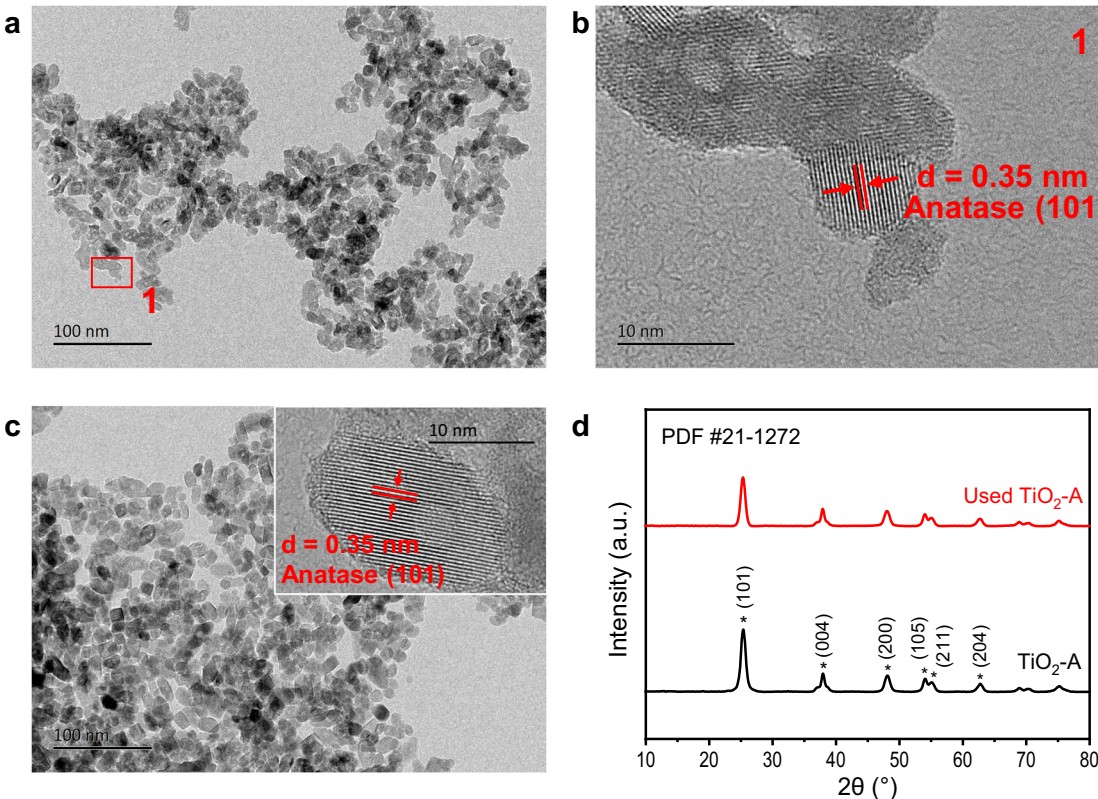

**Fig. 3 | Characterization of catalyst. a** and **b** TEM images of TiO$_2$-A; **c** TEM image of used TiO$_2$-A; **d** X-ray diffractograms of TiO$_2$-A and used TiO$_2$-A.

TiO$_2$-A. This may be due to the adsorption of phenol on the fresh TiO$_2$-A surface at the termination of the first reaction test and the surface has already reached saturated phenol adsorption in subsequent reuses of the TiO$_2$-A catalyst.

## Mechanism study

To study the reaction pathways, control experiments were carried out as illustrated in Fig. 4a. Since a trace amount of propylene and n-propyl ether was detected in the reaction, it is necessary to confirm whether they are intermediates or just by-products. No product was detected when 1-dodecene was used as the alkylation agent. With n-propyl ether, the yield and selectivity were not consistent with the results of using 1-propanol as reactant. In addition, we did not observe the high selectivity of n-propylphenol from Claisen rearrangement reaction of phenyl propyl ether on TiO$_2$-A, either. Therefore, we can rule out the possibility of alcohol firstly transforming into alkene or ether, or Claisen rearrangement reaction as a dominant process in the alkylation reaction.

The TiO$_2$-A was pretreated with 1-propanol to assess the influence of weak acidity by forming Alc-TiO$_2$-A (see the SI for details). Temperature-programmed desorption of NH$_3$ (NH$_3$-TPD) showed a significantly decreased amount of NH$_3$ adsorbed on Alc-TiO$_2$-A than on TiO$_2$-A, indicating a weaker acidity after the pretreatment (Fig. 4b). From the catalytic alkylation results, we found no significant change in the yield of 2-n-propylphenol, but the yield of isomerized product decreased (Fig. 4c). The result suggests that the production of isomerized product is due to the inherent acidity of TiO$_2$-A, while the α-C alkylation products may be attributed to other catalytic active sites. The Alc-TiO$_2$-A was further employed for the alkylation of phenol with 1-dodecanol, resulting in increased selectivity (87.1%) to superlinear α-C alkylation product at high phenol conversion (84.5%) as compared to TiO$_2$-A. The performance maintained after three-times reuse without additional treatment (Supplementary Fig. 3b), making it a superior catalyst for industrial process.

The role of oxygen vacancy on the TiO$_2$-A surface was studied to obtain preliminary insights on the catalytic sites for selective α-C alkylation. Our recent work showed that the surface of TiO$_2$-A can be partially reduced to generate oxygen vacancies by pretreating a TiO$_2$-A containing a very small amount of a transition metal (e.g. Au, Ag or Ni) under hydrogen at 400 °C[20–23]. Using Ni/TiO$_2$-A with 0.5 wt% Ni loading as the catalyst, phenol conversion was increased along with an increase of only α-C alkylation selectivity compared to fresh TiO$_2$-A (Fig. 4d). The result indicates that the oxygen vacancies on TiO$_2$-A surface were likely responsible for the generation of α-C alkylation products.

To probe the catalytic mechanism, experiment with deuterium substituted phenol as the substrate was carried out (Fig. 4e). After alkylation of phenol-$d_6$ with 1-propanol under standard conditions for 16 h, 2-n-propylphenol was separated by column chromatography. The $^2$H NMR spectrum showed deuterium signal at the α-position in the n-propyl group (Supplementary Fig. 5a). Combined with $^1$H NMR spectroscopic analysis of the product (Supplementary Fig. 5b), the percentage of deuterium on the α-position in n-propyl was determined in 45%. The result suggests that 90% of the isolated 2-n-propylphenol products had a deuterium atom and a hydrogen atom at the α-position of the n-propyl group. No deuterium signal was detected at the β-position and γ-position, which ruled out the possibility of H-D exchange at α-position. Therefore, we hypothesize that the deuterium atom at the *ortho* position of benzene ring may migrate from the aromatic C to the α-C of n-propyl group in the mechanism of alkylation during 1-propanol alkylation of phenol. The result clearly confirms that the mechanism of alkylation on TiO$_2$-A is fundamentally different from that of the known dominant electrophilic substitution reaction.

## Computational study

DFT calculations were carried out to help elucidate the α-C alkylation mechanism using 1-propanol as the representative. We firstly

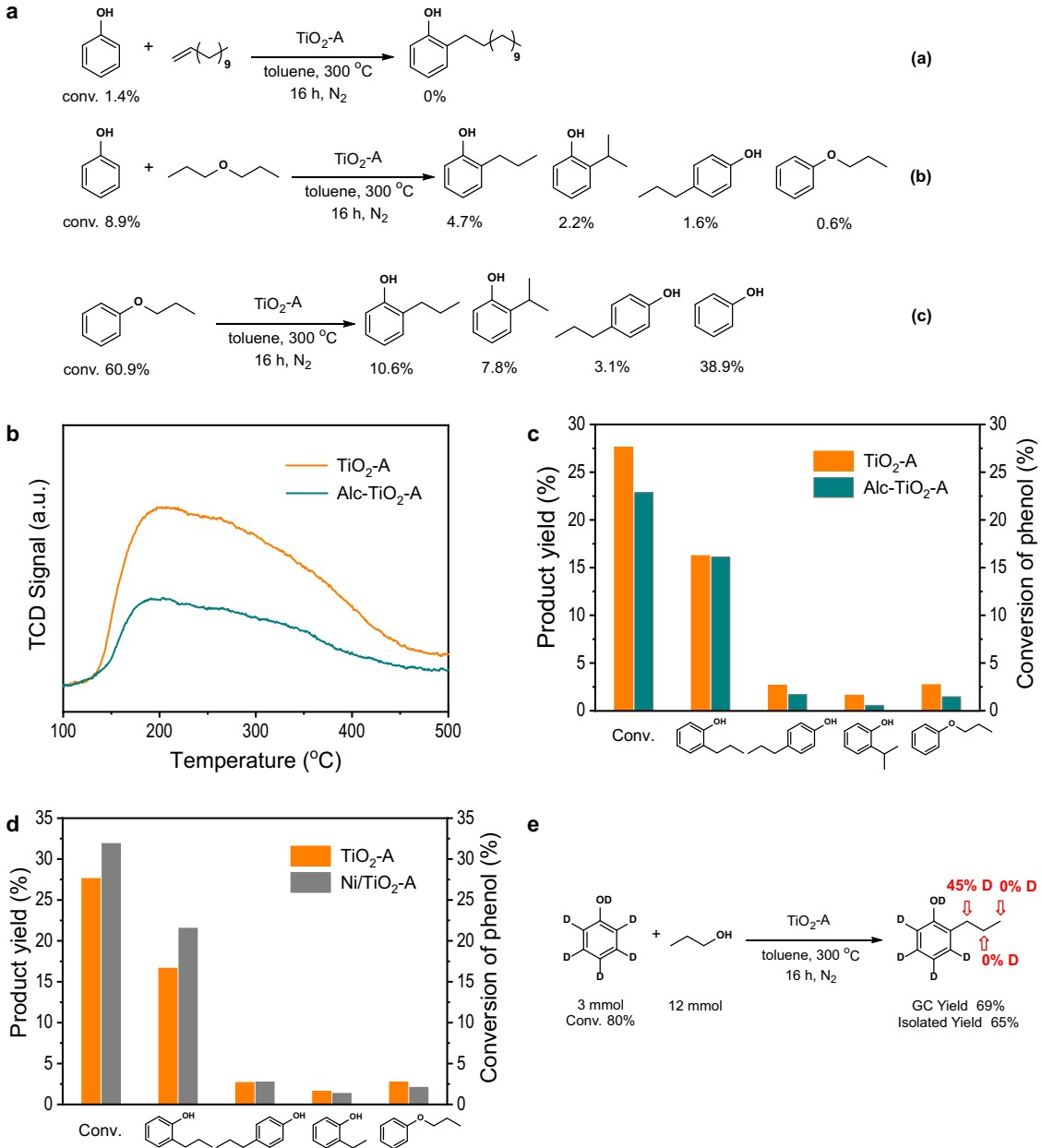

**Fig. 4 | Mechanism study. a** Control experiments; **b** NH$_3$-TPD profiles of TiO$_2$-A and Alc-TiO$_2$-A; **c** Product yields in the alkylation of phenol with 1-propanol over TiO$_2$-A and Alc-TiO$_2$-A; **d** Product yields in the alkylation of phenol with 1-propanol over TiO$_2$-A and Ni/TiO$_2$-A; **e** Alkylation of phenol-$d_6$ with 1-propanol. Reaction conditions in **c** and **d**: catalyst 0.2 g, phenol 5 mmol, 1-propanol 10 mmol, toluene 25 mL, 300 °C, 4 h, N$_2$ atmosphere. Ni loading of Ni/TiO$_2$-A is 0.5 wt%.

studied the distribution and diffusion of the oxygen vacancies that are indicated by experimental evidence to be crucial for the reactivity. It is known that on bare surface of TiO$_2$-A (101), the oxygen vacancy is more stable at subsurface than at surface sites[49–51], and it has been reproduced by our calculation. In addition, we find that with molecular or dissociated 1-propanol (or water) adsorbed, the oxygen vacancy is more stable at surface than at subsurface sites (by 0.26 eV for the molecular state and 0.42 eV for the C−H dissociated state, Supplementary Fig. 7). The barrier for vacancy diffusion is also quite small (0.29 eV in the case of 1-propanol adsorption), indicating that the substrate adsorption induces the diffusion of the vacancy from the subsurface to surface. Similar adsorption induced diffusion has also been observed for carboxylic acid group[52]. We propose this is because the vacancy on surface produces four coordinated Ti atom which is more unsaturated and can be favorably coordinated by substrates.

Based on the experimental and calculated results, we propose a reaction mechanism for the α-C alkylation featured by the titanium alkylidene intermediate (Ti=C bond). As shown in the blue curve in Fig. 5, firstly the C$_\alpha$−H activation reaches **chdiss** with the barrier of 1.88 eV. Then the isomerization of **chdiss** to a more stable isomer **chdiss_a** in which the hydroxyl group coordinates to the five coordinated Ti in [10$\bar{1}$] direction resulted from the surface vacancy. The Ti=C bond (**ti=c_a**) is readily formed through the reduction of the C−O bond by the electron polarons generated by the oxygen vacancy. The rate-determining step for the Ti=C bond formation is the C−H activation step (1.88 eV), which is slightly more difficult than the C−O breaking step (1.65 eV). Alternatively, **chdiss** can isomerize to **chdiss_b** in which the hydroxyl group coordinates to the surface five coordinated Ti in [010] direction, followed by a similar C−O addition to reach **ti=c_b** (red curve in Fig. 5). Our calculation shows this pathway has a much higher barrier than the **ti=c_a** pathway, but the barrier can be reduced by

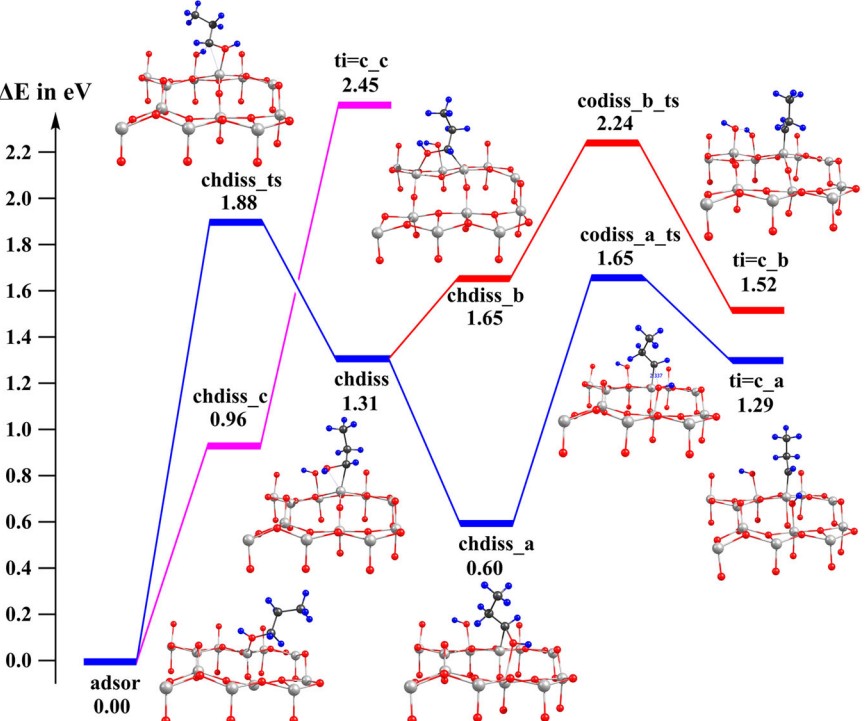

**Fig. 5 | The reaction path to generate Ti=C bond.** Blue curve: formation of Ti=C$_\alpha$ assisted by Ti5c generated by the surface vacancy; Red curve: formation of Ti=C$_\alpha$ assisted by surface Ti5c; Purple curve: formation of Ti=C$_\beta$. Black, blue, red and white spheres stand for C, H, O and Ti atoms, respectively.

introducing additional surface vacancies. Our mechanism also agrees with the n-propylphenol selectivity observed experimentally, since the yield of 2-isopropylphenol undergoes the Ti=C$_\beta$ intermediated **ti=c_c** which requires two strongly endothermic C–H activation, and is much more unstable than the Ti=C$_\alpha$ intermediated **ti=c_a** which only needs one C–H activation step (purple curve in Fig. 5).

Figure 6 (blue curve) shows the pathway for nucleophilic α-C alkylation of phenol at the *ortho* position by Ti=C bond. The dissociated state of phenol adsorbed adjacently to the titanium alkylidene. It undergoes the C–C bond formation to reach **ccform**, followed by hydrogen of the *ortho*-C migrates to the bridging oxygen (**hmig**), then migrates back to the propyl group to form the dissociated state of 2-n-propylphenol (**hback**). Finally, the adsorbed state of 2-n-propylphenol (**product**) is generated by proton transfer. All steps are fairly easy, with barriers of only 0.3 eV to 0.54 eV. We also studied the alkylation at the *meta* position (purple curve in Fig. 6). In line with the experimental results, we found its C–C bond formation is less facile than that at *ortho* position. The transition state (cc1-ts) is less stable by 0.32 eV and the key intermediate (cc1form) is less stable by 0.48 eV. Structural analysis shows that when the alkylidene and phenol co-adsorbed on the surface, the *meta*-C is much higher than the C(=Ti), while the *ortho*-C is only slightly higher. Therefore, the formation of the C–C bond at *ortho*-C needs less distortion and is easier. Finally, our mechanism matches the isotope experiments where the percentage of deuterium on the α-position in n-propyl was 45% when using phenol-$d_6$ as a probe for the alkylation, since one hydrogen of the α-position in n-propyl indeed comes from the hydrogen of phenol at *ortho* position. Certainly, there are more hydrogens on neighbor bridging oxygen (come from propanol), however, the hydrogen transfer between surface bridging oxygens (red curve in Fig. 6, barrier 0.73 eV from **hmig**) is more difficult than the proton transfer to form the product (0.43 eV from **hmig**), thus has little influence on the deuterium experiments. The desorption of the 2-n-propylphenol, together with H$_2$O forming (by adsorbed H and OH) and desorption, results in the bare surface with oxygen vacancy on the surface. The original catalyst TiO$_2$-A (101)

surface is then regenerated by the diffusion of the oxygen vacancy from the surface to the subsurface.

In short, we propose that the reaction proceeds via the key titanium alkylidene intermediate formed by C–H activation and C–O breaking of propanol, followed by the α-C alkylation of phenol and the proton transfer from phenyl to propyl through surface bridging oxygen (Fig. 7, and Supplementary Fig. 8 for details). The calculated rate-determining barrier, 1.88 eV, matches the experimental value 1.92 eV very well (Supplementary Fig. 6). Our mechanism also agrees with other experimental observations such as high selectivity to n-propylphenol, high selectivity to *ortho* alkylation, and the results of deuterium labeling experiment. Furthermore, our mechanism highlights the impact of oxygen vacancy since it offers low-coordinated Ti for better coordination of the substrate, and more importantly, yields electron polarons which reduce the C–O bond. It also shows that the titanium alkylidene, previously reported in a homogenous system and stabilized by sterically bulky ligands[53,54], can be generated on alcohol-adsorbed TiO$_2$-A surface in situ, and show remarkable reactivities.

In conclusion, we find that pure TiO$_2$-A alone catalyzes the nucleophilic alkylation of phenols with alcohols to produce α-C alkylation products. The catalyst exhibits excellent selectivity and stability, making it sustainable and scalable for the direct synthesis of linear alkylphenols. The formation of Ti=C– bond with the α-carbon of the alkyl group at oxygen vacancies is reported for the first time on a metal-free TiO$_2$-A surface. Insights on the active site and α-C alkylation mechanism are demonstrated for the selective C–C bond formation.

## Methods
### Chemicals and catalysts
Rutile TiO$_2$ (20 nm, 99.9%, labeled TiO$_2$-R), P25 TiO$_2$ (20 nm, 99.9%), γ-Al$_2$O$_3$ (20 nm, 99.9%), and all organic chemicals were purchased from Aladdin Industrial Co. Ltd without further purifications. Commercial H-ZSM-5 with a nSiO$_2$/nAl$_2$O$_3$ ratio of 25 was purchased from Nankai University Catalyst Co. Ltd.

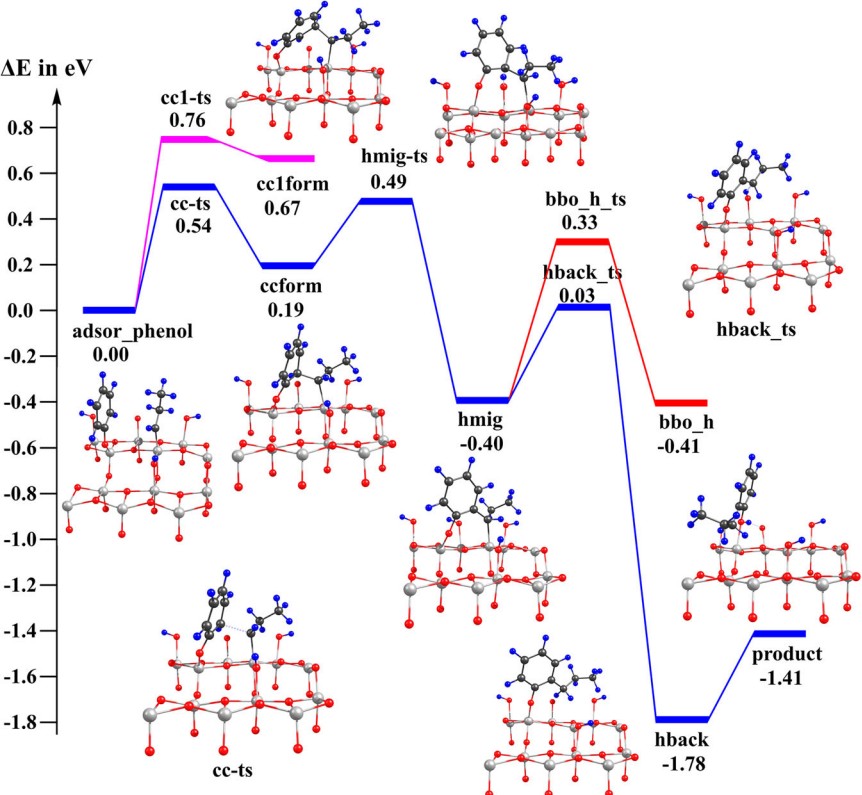

**Fig. 6 | The reaction path for the nucleophilic α-C alkylation of phenol by Ti=C bond.** Blue curve: alkylation at *ortho* position; Purple curve: alkylation at *meta* position; Red curve: hydrogen diffusion along surface bridging oxygen. Black, blue, red and white spheres stand for C, H, O and Ti atoms, respectively.

Anatase $TiO_2$ (labeled $TiO_2$-A) was prepared by the sol-gel method. 20 mL tetra butyl titanate was dissolved in 100 mL of ethanol to form solution A. 20 mL water, 20 mL ethanol, and 12 mL acetic acid were mixed to form solution B. Under vigorous magnetic stirring, solution B was added to solution A dropwise and kept stirring for 12 hours. The gel was placed for another 12 hours and washed with ultrapure water and separated by centrifugation. The sample was dried overnight at 110 °C after washing it five times. Finally, it was heated to 400 °C at a rate of 2 °C/min in a muffle furnace and kept for 4 h to obtain $TiO_2$-A.

Ni-loaded anatase $TiO_2$ (labeled Ni/$TiO_2$-A) was prepared by an incipient wetness impregnation method with an aqueous solution of Ni(NO$_3$)$_2$. The sample was dried overnight at 120 °C, and then heated to 400 °C at a rate of 10 °C/ min and kept at 400 °C for 4 h. The calcined sample was further kept at 400 °C for 1 h in 10 vol% $H_2$/Ar and then cooled to room temperature under $N_2$ atmosphere.

## Characterization

Transmission electron micrographs (TEM) were obtained on a JEM-2100 microscope operated at 200 kV. The samples were suspended in ethanol and a few drops of the suspension were dried to the TEM grid for TEM measurement. X-ray diffractograms of the samples were obtained on a PANalytical XPert Powder X-ray diffractometer with a Cu Kα radiation. The measurement was operated at 40 kV and scanning 2θ from 5° to 80° with a step of 0.013°. The signal was collected by a pixel 1D detector, and the data were analyzed by comparison with reference patterns in the database (PDF2-2004). X-ray photoelectron spectra (XPS) were recorded on a Thermo Scientific K-Alpha equipped with a monochromatic Al Kα X-ray radiation source ($hv$ = 1486.6 eV). The C 1s peak was used as the reference at 284.8 eV. The specific surface area was measured on a Micromeritics ASAP 2020 physical adsorption analyzer. The measurement was operated with the Brunauer-Emmett-Teller (BET) method using $N_2$ adsorption at 77.3 K. The sample was

degassed at 200 °C for 5 h before the measurement. NH$_3$ temperature-programmed desorption (NH$_3$-TPD) was performed on a Micromeritics AutoChem II 2920 chemisorption analyzer. The catalyst (150 mg) was refreshed for 60 min under an argon (Ar) atmosphere at 400 °C, cooled to 100 °C, and then saturated for 60 min with 10 vol% NH$_3$/He. After that, the catalyst was flushed in a He flow for 60 min at 100 °C. Finally, the NH$_3$-TPD was executed by heating the catalysts in He (10 °C/min) from 100 to 400 °C. The desorbed NH$_3$ was detected with a thermal conductivity detector (TCD). NMR was performed on a Bruker AVANCE III 400 spectrometer instrument in deuterated chloroform (CDCl$_3$). Chemical shifts are reported in parts per million (ppm) downfield from TMS.

## Catalytic evaluation

Alkylation reactions were carried out in a 50 mL stainless steel batch reactor, from Beijing Century Senlong Experimental Apparatus Co. Ltd. The reactor was equipped with a mechanical stirrer, a thermo-couple, a pressure gauge, and a programmable controller. In a typical run, 5 mmol of phenol, 10 mmol of 1-propanol, 25 mL of toluene, and 0.2 g of catalyst were loaded into the reactor. The reactor was purged with $N_2$ for 10 min to remove the air and get to a pressure of 1 MPa. The reactor was then heated to 300 °C and kept for a specified reaction time while the content was stirred at a rate of 500 rpm. After the reaction, 40.2 mg of n-hexadecane as an internal standard and 25 mL of ethanol were added into the reactor. The reaction products were identified by GC-MS (Agilent 7890A-5975C, HP-5MS) and quantified by GC (Agilent 7890 A) with a flame ionization detector (FID) using an HP-5 column (30 m × 0.32 mm × 0.25 μm).

The conversion of phenol, selectivity of phenolic compounds and product yield are calculated by Eqs. 1, 2 and 3, respectively. In these equations, $n_{\text{initial phenol}}$ and $n_{\text{final phenol}}$ are the molar amount of phenol before and after reaction. $n_{\text{product i}}$ is the molar amount of aromatic

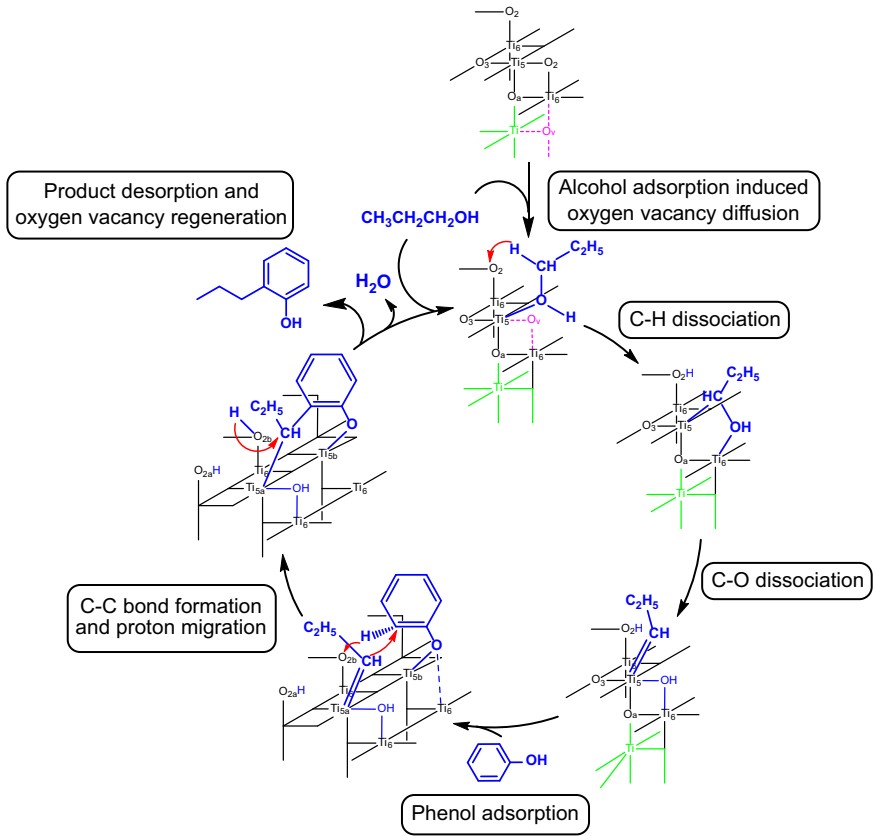

**Fig. 7 | Schematic overview of the mechanism for nucleophilic α-C alkylation on TiO₂-A.** The alcohol adsorption induces the diffusion of oxygen vacancy from subsurface to surface on TiO₂-A. It undergoes C–H activation and C–O dissociation to form Ti=Cα intermediate. Then nucleophilic α-C alkylation reacts at the *ortho* position of phenol, followed by the proton transfer through surface bridging oxygen. Finally, the oxygen vacancy is regenerated by product desorption.

product i in the reaction mixture.

$$\text{conversion}(\%) = \frac{n_{\text{initial phenol}} - n_{\text{final phenol}}}{n_{\text{initial phenol}}} \times 100\% \quad (1)$$

$$\text{selectivity}(\%) = \frac{n_{\text{product }i}}{n_{\text{initial phenol}} - n_{\text{final phenol}}} \times 100\% \quad (2)$$

$$\text{yield }(\%) = \frac{n_{\text{product }i}}{n_{\text{initial phenol}}} \times 100\% \quad (3)$$

We also studied the molar mass balance of alcohols. The conversion of alcohol and product yield are calculated by Eq. 4 and Eq. 5. In these equations, $n_{\text{initial alcohol}}$ and $n_{\text{final alcohol}}$ are the molar amount of alcohol before and after reaction. $n_{\text{product }i}$ is the molar amount of mono alkyl group in product i after the reaction.

$$\text{conversion}(\%) = \frac{n_{\text{initial alcohol}} - n_{\text{final alcohol}}}{n_{\text{initial alcohol}}} \times 100\% \quad (4)$$

$$\text{yield}(\%) = \frac{n_{\text{product }i}}{n_{\text{initial alcohol}}} \times 100\% \quad (5)$$

## Computational details
All calculations were performed with PBE functional[55] using the Vienna ab initio simulation package code[56] and plane augmented wave potential[57]. The wave function was expanded by the plane wave, with a kinetic cut-off of 400 eV and density cut-off of 650 eV.

An efficient force reversed method[58] was used to locate the transition state (TS). Our surface model was cut out of a six-layer slab anatase TiO₂ crystal to expose the (101) surface. The periodically repeated slabs on the surface were decoupled by 15 Å vacuum gaps. A Monkhorst−Pack grid[59] of single k-points was used for the 5 × 2 surface unit cell. The vdW-dispersion energy was corrected by DFT-D3 method of Grimme et al.[60].

## Data availability
The data supporting the findings of this study are available in the paper and its Supplementary Information. All other data are available from the corresponding authors upon reasonable request. Source data are provided with this paper.

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

## Acknowledgements
This work was supported by the National Natural Science Foundation of China (Grants 21932005, 22172164, 92061114, 21873096).

## Author contributions
Z.C.Z. conceived the study and guided the project. X.Z.D. performed the experiments and characterizations. H.J.F. conducted DFT calculations. S.L.L. participated in the discussion. All the authors contributed to the writing of the manuscript.

## Competing interests
The authors declare no competing interests.
