## [Peer Review File · Nature Communications]

REVIEWER COMMENTS

Reviewer #1 (Remarks to the Author):

Du et al. report a selective nucleophilic alkylation of phenols with primary alcohols via Ti=C α intermediate on anatase TiO₂ surface. The catalytic performance is promising and the mechanism is investigated by deuterium exchanged substrate and DFT calculations. Overall, it is a piece of interesting work to be published on Nature Communications. However, before I can recommend its acceptance, the following issues should be addressed.

1. The authors should give the calculation methods for the product yield and the carbon balance.
2. The boiling points of polyalkylation products are generally very high. Is GC-MS accurate for quantitative detection?
3. The feeding ratio between phenol and 1-propanol is 1:4, why? Does the ratio significantly influence the catalytic performance?
4. The authors explore the variations in phenol conversion in various solvent systems and select toluene as the optimal solvent. Will it participate in the reaction and compete with phenol?
5. To demonstrate that the crystal structure did not change appreciably during the reaction, high resolution electron microscopic images, such as those in Fig. 2b, should be presented.
6. The characterizations of oxygen vacancies should be added. For example, O1s XPS, EPR. Besides, it will be more convincing if the authors can present the semi-quantitative relationship between catalytic performance and the content of oxygen vacancies.
7. Lines 235-237, "We propose this is because the vacancy on surface produces four coordinated Ti atom which is more unsaturated and can be favorably coordinated by substrates." Do Ti XPS data support this hypothesis?
8. Most importantly, the proposed mechanism is lacking experimental evidence, particularly for in situ spectroscopy of the Ti=C α intermediate.

Reviewer #2 (Remarks to the Author):

This work by Fan and Zhang describes an anatase TiO₂-catalyzed nucleophilic alkylation of phenols with alcohols to produce linear n-alkylphenols. This reaction proceeds with decent yield as well as

good regioselectivity. This work builds on a fairly extensive body of literature on ortho-selective C–H functionalization reactions of free phenols with cheap, sustainable and scalable solid catalyst. Mechanistic and computational studies suggest that the reaction proceeds via a titanium alkylidene intermediate followed by the n-alkylation of phenol and the proton transfer from phenyl to propyl through surface bridging oxygen. Overall, this is a nice contribution to the formation of Ti=C– bond that expands the precursor scope to include primary alcohol on a metal free TiO₂-A surface. Important literature precedents are clearly cited, and the scholarship of the study is high. I therefore support publication in Nat. Commun. pending the minor revisions

1. What about the results with secondary alcohols?
2. The scope of phenols is inadequate. More functional groups need to be tested.
3. What about the results with naphthols?
4. What about the reusability of the TiO₂-A catalyst?

Response to Reviewers

Response to Reviewer #1

Comment (1): The authors should give the calculation methods for the product yield and the carbon balance.

Response: Thank you for your comments. We have added the calculation methods and the marked sentences in the Methods section of the manuscript as “The conversion of phenol, selectivity of phenolic compounds and product yield are calculated by Eq. 1, Eq. 2 and Eq. 3, respectively. In these equations, $n_{\text{initial phenol}}$ and $n_{\text{final phenol}}$ are the molar amount of phenol before and after reaction. $n_{\text{product i}}$ is the molar amount of aromatic product i in the reaction mixture.

$$\text{conversion (\%)} = \frac{n_{\text{initial phenol}} - n_{\text{final phenol}}}{n_{\text{initial phenol}}} \times 100\% \quad (1)$$

$$\text{selectivity (\%)} = \frac{n_{\text{product i}}}{n_{\text{initial phenol}} - n_{\text{final phenol}}} \times 100\% \quad (2)$$

$$\text{yield (\%)} = \frac{n_{\text{product i}}}{n_{\text{initial phenol}}} \times 100\% \quad (3)$$

We also studied the molar mass balance of alcohols. The conversion of alcohol and product yield are calculated by Eq. 4 and Eq. 5. In these equations, $n_{\text{initial alcohol}}$ and $n_{\text{final alcohol}}$ are the molar amount of alcohol before and after reaction. $n_{\text{product i}}$ is the molar amount of mono alkyl group in product i after the reaction.”

$$\text{conversion (\%)} = \frac{n_{\text{initial alcohol}} - n_{\text{final alcohol}}}{n_{\text{initial alcohol}}} \times 100\% \quad (4)$$

$$\text{yield (\%)} = \frac{n_{\text{product i}}}{n_{\text{initial alcohol}}} \times 100\% \quad (5)$$

Comment (2): The boiling points of polyalkylation products are generally very high. Is GC-MS accurate for quantitative detection.

Response: Thank you for your comments. The polyalkylation products generated by solid acid catalysts were identified by GC-MS, and calculated by the molar balance of phenol in our previous version of manuscript. In the revised manuscript, we

recalculate the yield of polyalkylation products by ^1H NMR spectra with an internal standard method. By combining results of GC-FID and NMR methods, all alkylation products are accurately quantified now. We have updated the results in Fig. 1 and added the marked sentence in the manuscript as “Several isopropyl polyalkylation products are identified by GC-MS and calculated by NMR spectra with the internal standard method.”

Comment (3): The feeding ratio between phenol and 1-propanol is 1:4, why? Does the ratio significantly influence the catalytic performance?

Response: Thank you for your comments. As shown in Fig. S1C in the revised SI, we examined the feeding ratio between phenol and 1-propanol under standard reaction conditions. With the feeding ratio varying from 1:1 to 1:4, product selectivity had no obvious changes, and conversions of phenol gradually increased. This is due to the relatively high concentration of 1-propanol favors the reaction equilibrium. But the catalytic nature of $\text{TiO}_2\text{-A}$ is not affected by the feeding ratio. We have added the results as Fig. S1C in the revised SI and the marked sentence in the manuscript as “With the feeding ratio between phenol and 1-propanol varying from 1:1 to 1:4, product selectivity had no obvious changes while conversions of phenol gradually increased (Fig. S1C) as a relatively high concentration of 1-propanol favorably drives the reaction equilibrium to the product.”

Comment (4): The authors explore the variations in phenol conversion in various solvent systems and select toluene as the optimal solvent. Will it participate in the reaction and compete with phenol?

Response: Thank you for your comments. As shown in the following Scheme R1, we examined several arenes to react with 1-propanol under standard reaction conditions. No product was detected, and therefore they would not participate in the reaction when used as the solvent.

Scheme R1 Catalytic reaction of arenes with 1-propanol. Reaction conditions: TiO₂-A 0.2 g, 1-propanol 10 mmol, arene 25 ml, 300 °C, 16h, N₂.

Comment (5): To demonstrate that the crystal structure did not change appreciably during the reaction, high-resolution electron microscopic images, such as those in Fig. 2b, should be presented.

Response: Thank you for your suggestion. **The HRTEM image of used TiO₂-A was added in Fig. 2c in the manuscript.** The crystal structure of TiO₂-A did not change obviously after the reaction.

Comment (6): The characterizations of oxygen vacancies should be added. For example, O 1s XPS, EPR. Besides, it will be more convincing if the authors can present the semi-quantitative relationship between catalytic performance and the content of oxygen vacancies.

Response: Thank you for your suggestion. The characterizations of oxygen vacancy of TiO₂ have been widely reported using XPS and EPR. In these works, the oxygen vacancy-rich TiO₂ is synthesized by specific methods, such as hydrogen treatment,^{1,2} solvothermal reaction,^{3,4} anion doping,⁵ and so on. However, in our work, the pristine TiO₂-A was used as the catalyst, and the computational studies showed that diffusion of oxygen vacancy from the subsurface to surface was induced by substrate adsorption in the reaction. Therefore, in situ characterization methods are essential for the detection of surface oxygen vacancy of TiO₂-A. Unfortunately, it is still challenging to conduct in situ XPS or EPR tests with substrate feeding at 300 °C.

We also prepared a propanol adsorbed TiO₂-A sample in the glovebox under Ar

atmosphere and performed the XPS and EPR tests at room temperature. As shown in Fig. R1, the valence state of O can be divided into two parts, lattice oxygen (529.7 eV) and hydroxyl oxygen (531.0 eV) on the surface. Compared with bare TiO₂-A, the O 1s XPS spectra of propanol adsorbed TiO₂-A showed no obvious shift. Fig. R2 shows the EPR spectra of bare TiO₂-A and propanol adsorbed TiO₂-A. No signals of surface oxygen vacancy were detected for these two samples.

Fig. R1 The O 1s XPS spectra of bare TiO₂-A and propanol adsorbed TiO₂-A.

Fig. R2 EPR spectra of bare TiO₂-A and propanol adsorbed TiO₂-A.

Comment (7): Lines 235-237, “We propose this is because the vacancy on surface produces four coordinated Ti atom which is more unsaturated and can be favorably coordinated by substrates.” Do Ti XPS data support this hypothesis?

Response: Thank you for your comments. As discussed above, the surface oxygen vacancy is induced by substrate adsorption, and then produces four coordinated Ti

atom. The unsaturated Ti atom is favorable to coordinate with substrate and then involved in the subsequent reaction process quickly. The operando XPS test is necessary to monitor the valance state of Ti of TiO₂-A surface during the reaction, but this technology is still limited now. Fig. R3 shows the Ti 2p XPS spectra of bare TiO₂-A and propanol adsorbed TiO₂-A at room temperature. And the peak shows no obvious shift in this ex-situ test.

Fig. R3 The Ti 2p XPS spectra of bare TiO₂-A and propanol adsorbed TiO₂-A.

Comment (8): Most importantly, the proposed mechanism is lacking experimental evidence, particularly for in situ spectroscopy of the Ti=C_α intermediate.

Response: Thank you for your comments. Certainly, the experimental evidence of Ti=C_α intermediate is vital for the alkylation mechanism we proposed. The isotope experiments verify the alkylidene intermediate in reaction, which provides indirect evidence for Ti=C_α intermediate. In situ IR spectroscopy may be a potential characterization method. As shown in Fig. R4, we conducted the in situ IR test of propanol adsorbed TiO₂-A sample under Ar atmosphere with bare TiO₂-A as background. The signals of propanol molecule indicate the existence of adsorbed propanol on the surface of TiO₂-A even at 300 °C. However, with the temperature increasing to 300 °C, no extra signals were observed obviously within the range of 600 to 4000 cm⁻¹. Although the IR signals of Ti=C bond have been previously reported, these titanium alkylidenes are stable complexes with sterically bulky ligands.⁶⁻¹³ In our work, the titanium alkylidene intermediate was generated on

TiO₂-A surface in situ. The short lifetime and small amount of the alkylidene intermediate make direct characterization infeasible for unambiguous quantification. And it is still challenging now to detect because of the technical limitation.

Fig. R4 In situ IR spectra of propanol adsorbed TiO₂-A.

References

1. Chen, X., Liu, L., Yu, P. Y. & Mao, S. S. Increasing solar absorption for photocatalysis with black hydrogenated titanium dioxide nanocrystals. *Science* **331**, 746-750 (2011).
2. Chen, X., Liu, L. & Huang, F. Black titanium dioxide (TiO₂) nanomaterials. *Chem. Soc. Rev.* **44**, 1861-1885 (2015).
3. Bi, X. *et al.* Tuning oxygen vacancy content in TiO₂ nanoparticles to enhance the photocatalytic performance. *Chemical Engineering Science* **234** (2021).
4. Benavides-Guerrero, J. A. *et al.* Synthesis of vacancy-rich titania particles suitable for the additive manufacturing of ceramics. *Sci Rep* **12**, 15441 (2022).
5. Fittipaldi, M., Gatteschi, D. & Fornasiero, P. The power of EPR techniques in revealing active sites in heterogeneous photocatalysis: The case of anion doped TiO₂. *Catal. Today* **206**, 2-11 (2013).
6. Cundari, T. R. & Gordon, M. S. High-valent transition-metal alkylidene complexes: effect of ligand and substituent modification. *J. Am. Chem. Soc.* **114**, 539-548 (2002).
7. Bailey, B. C. *et al.* Intermolecular C-H bond activation promoted by a titanium alkylidyne. *J. Am. Chem. Soc.* **127**, 16016-16017 (2005).
8. Basuli, F. *et al.* Four-Coordinate Titanium Alkylidene Complexes: Synthesis, Reactivity, and Kinetic Studies Involving the Terminal Neopentylidene Functionality. *Organometallics* **24**, 1886-1906 (2005).
9. Andrews, L. & Cho, H.-G. Matrix Preparation and Spectroscopic and Theoretical Investigations of Simple Methylidene and Methylidyne Complexes of Group 4-6 Transition Metals. *Organometallics* **25**, 4040-4053 (2006).
10. Lyon, J. T. & Andrews, L. An Infrared Spectroscopic and Theoretical Study of Group 4 Transition Metal CH₂MCl₂ and HC≡MCl₃ Complexes. *Organometallics* **26**, 332-339 (2006).
11. Bailey, B. C., Fan, H., Huffman, J. C., Baik, M. H. & Mindiola, D. J. Intermolecular C-H bond

activation reactions promoted by transient titanium alkylidynes. Synthesis, reactivity, kinetic, and theoretical studies of the Ti[triple bond]C linkage. *J. Am. Chem. Soc.* **129**, 8781-8793 (2007).

12. Lyon, J. T. & Andrews, L. Group 4 transition metal CH₂=MF₂, CHF=MF₂, and HC/MF₃ complexes formed by C-F activation and alpha-fluorine transfer. *Inorg Chem* **46**, 4799-4808 (2007).

13. Svitova, A. L. *et al.* Endohedral fullerene with mu₃-carbido ligand and titanium-carbon double bond stabilized inside a carbon cage. *Nat. Commun.* **5**, 3568 (2014).

Response to Reviewer #2

Comment (1): What about the results with secondary alcohols?

Response: Thank you for your comments. As shown in Scheme 1 in the revised manuscript, two secondary alcohols, 2-propanol (Entry 3) and 2-dodecanol (Entry 4) were employed to react with phenol under standard reaction conditions. The conversions of phenol with these two secondary alcohols are similar to the corresponding primary alcohols. And the main products suggest it still maintains the α -C alkylation process. The results with secondary alcohols are also consistent with our proposed mechanism. We have added the results in Scheme 1 and the marked sentences in the manuscript as “In addition, secondary alcohols, such as 2-propanol and 2-dodecanol, were also employed to react with phenol (Scheme 1, entries 3 and 4). The TiO₂-A catalyst exhibited similar reactivity to the corresponding primary alcohols, and more importantly, maintained the selectivity of α -C alkylation.”

Comment (2): The scope of phenols is inadequate. More functional groups need to be tested.

Response: Thank you for your suggestion. As shown in Scheme S1 in the revised SI, besides the 3-chlorophenol and 3-methylphenol in the previous manuscript, more substituent phenols were tested, including amino group, nitro group, methoxy group and dimethyl group as substituted groups at *meta* position of phenol (Scheme S1, Entry 1-4). Generally, the conversions of phenols are higher than 84%, with more than 80% selectivity to *ortho*-substituted α -C alkylation products. The effect of substituent groups on catalytic performance is slight. In addition, 4-chlorophenol and 4-methylphenol (Scheme S1, Entry 5-6) were tested to study the effect of substituent position. The *para* substituted phenols also exhibited good reactivity. We have added the results in Scheme S1 and the marked sentence in the manuscript as “Phenols with additional substituents were further tested by reacting with 1-propanol (Scheme S1, entries 1-6), and high conversions of phenols (more than 84%) and selectivity to *ortho*-substituted n-propyl products (more than 80%) were obtained.”

Comment (3): What about the results with naphthols?

Response: Thank you for your comments. As shown in Scheme S1 in the revised SI, we tested several naphthols to react with 1-propanol under standard reaction conditions (Scheme S1, Entry 7-10). The 1-naphthol conversion rate reached 93.6%, with 87.9% selectivity to 2-n-propyl-1-naphthol. Substituted 1-naphthols with methyl, chlorine and methoxy groups were also studied and more than 90% conversions were obtained. It is worth noting that the selective alkylation at *ortho* position of hydroxyl group was still maintained for naphthols. Overall, the substrate scope of our TiO₂-A catalyzed selective alkylation process can be extended to naphthols. **We have added the results in Scheme S1 and the marked sentences in the manuscript as “In addition, naphthols as alternative substrates also showed similar alkylation selectivity in high conversions (Scheme S1, entries 7-10). These results demonstrate that the TiO₂-A catalyzed selective α -C alkylation reaction applies to a broad scope of substrates.”**

Comment (4): What about the reusability of the TiO₂-A catalyst?

Response: Thank you for your comments. As shown in Fig. S2A in the revised SI, we studied the reusability of TiO₂-A in alkylation reaction of phenol with 1-propanol. After three-times reuse of the TiO₂-A, the phenol conversion and product yields remained no obvious changes. The other example of reuse test is the alkylation reaction of phenol with 1-dodecanol (Fig. S2B). The improved Alc-TiO₂-A catalysts also showed good catalytic stability. **We have marked the sentences in the manuscript as “Its structure stability was maintained with multiple reuse tests. After three-times reuse of the TiO₂-A, the phenol conversion and product yields remained no obvious changes (Fig. S2A). It is interesting to note that the used TiO₂-A had a better mass balance than the fresh TiO₂-A. This may be due to the adsorption of phenol on the fresh TiO₂-A surface at the termination of the first reaction test and the surface has already reached saturated phenol adsorption in subsequent reuses of the TiO₂-A catalyst.” “The performance maintained after three-times reuse without additional treatment (Fig. S2B), making it a superior catalyst for industrial process.”**

REVIEWERS' COMMENTS

Reviewer #2 (Remarks to the Author):

I have reviewed an earlier version of this manuscript and the authors have adequately addressed my previous comments as well as those of the other reviewers. I therefore support publication in Nat. Commun. with current version.

The point-to-point responses to the comments by reviewers are given below:

Response to Reviewer #2

Comment: I have reviewed an earlier version of this manuscript and the authors have adequately addressed my previous comments as well as those of the other reviewers. I therefore support publication in Nat. Commun. with current version.

Response: Thank you very much for your valuable comments. The professional suggestions have enabled us to greatly improve the manuscript.